# Minimally Modified HIV-1 Infection of Macaques: Development, Utility, and Limitations of Current Models

**DOI:** 10.3390/v16101618

**Published:** 2024-10-16

**Authors:** Manish Sharma, Mukta Nag, Gregory Q. Del Prete

**Affiliations:** AIDS and Cancer Virus Program, Frederick National Laboratory for Cancer Research, Frederick, MD 21702, USA; manish.sharma@nih.gov (M.S.); mukta.nag@nih.gov (M.N.)

**Keywords:** HIV-1, AIDS, SIV, animal model, nonhuman primate, macaque

## Abstract

Nonhuman primate (NHP) studies that utilize simian immunodeficiency virus (SIV) to model human immunodeficiency virus (HIV-1) infection have proven to be powerful, highly informative research tools. However, there are substantial differences between SIV and HIV-1. Accordingly, there are numerous research questions for which SIV-based models are not well suited, including studies of certain aspects of basic HIV-1 biology, and pre-clinical evaluations of many proposed HIV-1 treatment, prevention, and vaccination strategies. To overcome these limitations of NHP models of HIV-1 infection, several groups have pursued the derivation of a minimally modified HIV-1 (mmHIV-1) capable of establishing pathogenic infection in macaques that authentically recapitulates key features of HIV-1 in humans. These efforts have focused on three complementary objectives: (1) engineering HIV-1 to circumvent species-specific cellular restriction factors that otherwise potently inhibit HIV-1 in macaques, (2) introduction of a C chemokine receptor type 5 (CCR5)-tropic envelope, ideally that can efficiently engage macaque CD4, and (3) correction of gene expression defects inadvertently introduced during viral genome manipulations. While some progress has been made toward development of mmHIV-1 variants for use in each of the three macaque species (pigtail, cynomolgus, and rhesus), model development progress has been most promising in pigtail macaques (PTMs), which do not express an HIV-1-restricting tripartite motif-containing protein 5 α (TRIM5α). In our work, we have derived a CCR5-tropic mmHIV-1 clone designated stHIV-A19 that comprises 94% HIV-1 genome sequence and replicates to high acute-phase titers in PTMs. In animals treated with a cell-depleting CD8α antibody at the time of infection, stHIV-A19 maintains chronically elevated plasma viral loads with progressive CD4+ T-cell loss and the development of acquired immune-deficiency syndrome (AIDS)-defining clinical endpoints. However, in the absence of CD8α+ cell depletion, no mmHIV-1 model has yet displayed high levels of chronic viremia or AIDS-like pathogenesis. Here, we review mmHIV-1 development approaches, the phenotypes, features, limitations, and potential utility of currently available mmHIV-1s, and propose future directions to further advance these models.

## 1. Introduction

Productive human immunodeficiency virus type 1 (HIV-1) infection is restricted to only three or four animal species: humans, chimpanzees, gibbon apes, and perhaps gorillas [1,2,3,4]. Thus, nonhuman primate (NHP) models of HIV-1 infection have historically utilized simian immunodeficiency viruses (SIVs), which are related to but distinct from HIV-1, or chimeric viruses primarily composed of SIV sequence (simian-human immunodeficiency viruses, or SHIVs). Typically, these viruses are experimentally inoculated into one of three Asian macaque species: rhesus macaques (RM; *Macaca mulatta*), cynomolgus macaques (CM; *Macaca fascicularis)*, or pigtail macaques (PTM; *Macaca nemestrina*). With the right combination of virus and host macaque species, many of the key features of HIV-1 infection in humans can be recapitulated in these models, including predominant infection of CD4^+^ T cells [5,6,7], early destruction of CD4^+^ T cells in gut mucosal sites [7,8,9], the development of chronic immune activation and inflammation [10,11,12,13], the establishment of a persistent rebound-competent viral reservoir [14,15,16], and the gradual depletion of CD4^+^ T cells in blood and progression to acquired immunodeficiency syndrome (AIDS) [17,18]. While these models have been invaluable for advancing our understanding of HIV-1 pathogenesis, and providing systems in which to test prevention, treatment, and cure strategies, there are key differences between SIV/SHIV and HIV-1 that have motivated efforts to develop NHP models that utilize viruses that are more genetically similar to HIV-1. We and several other groups have pursued the development of a minimally modified HIV-1 (mmHIV-1) that can replicate to high titers in macaques and can recapitulate the key features of HIV-1 infection and pathogenesis in humans. In this review, we provide the rationale for generating mmHIV-1-based models, summarize the status of these efforts to date including the strengths and limitations of current models, and discuss potential future work toward the further characterization, evaluation, and establishment of these mmHIV-1-based NHP models of HIV-1 infection.

## 2. Rationale for the Development of Minimally Modified HIV-1-Based NHP Models

There are several important differences between HIV-1 and the SIVsmm and SIVmac strains used in pathogenic NHP models of HIV-1 infection. Genetically, the HIV-1 and SIV genomes share only approximately 50% nucleotide similarity and differ in the arrangement of the overlapping regions of their open reading frames [19,20]. Moreover, these SIV strains do not possess a *vpu* gene as HIV-1 does, but instead have a different accessory gene, *vpx*, that conversely is not present in HIV-1. These genetic differences can result in substantial differences in the structure, function and antigenicity between the HIV-1 and SIV viral proteins, precluding the use of SIV to directly evaluate the effectiveness of HIV-specific vaccines/immunogens or treatments and other interventions that differ in their potency against SIV compared with HIV.

Simian-human immunodeficiency viruses, engineered chimeric viruses comprising HIV-1 gene regions swapped into an otherwise SIV genome, were developed to overcome some of the limitations of SIV models. Early SHIV development involved replacement of the SIV *env* gene and flanking regions with the corresponding HIV-1 sequences to enable the evaluation of HIV-1 Env-specific interventions and immunity, particularly antibodies, including both passively administered antibodies and antibodies elicited through immunization, in NHP models. Since the earliest described Env-SHIVs, SHIV-based NHP models have significantly improved through the generation and in vivo evaluation of SHIVs with relevant coreceptor tropism [21,22], different neutralization sensitivity profiles [23,24,25], clinically relevant transmitted/founder Envs [26,27], and the identification of specific point-mutations that can enhance the ability of HIV-1 Envs to engage the macaque CD4 receptor molecule, enabling a broader range of HIV-1 envelopes to be effectively utilized in SHIV models [26,28,29].

While SHIV-based models have proven useful for studies involving Env-targeted interventions, they do not provide an avenue for evaluating HIV-specific immune responses or vaccination approaches directed toward other viral gene products. In addition, many antiretroviral drugs (ARVs) developed to inhibit HIV-1 replication have reduced potency against SIV and SHIVs alike. While in some cases this reduced potency can be overcome by simple drug dosage adjustments, for other agents drug metabolism and toxicity in macaques, drug formulation requirements and availability, and the extent of reduced potency against SIV/SHIV preclude this approach. It also remains unclear how the complex, often non-linear relationship between drug concentrations in blood and in key tissue sites might impact the assessed activity of a new agent against viruses with substantially lower sensitivity to the agent than circulating HIV-1. Non-nucleoside reverse transcriptase inhibitors (NNRTIs), which have dramatically reduced potency against reverse transcriptase (RT) enzymes encoded by SIV compared with HIV-1 [30], represent the prototypical example of an entire class of clinically relevant ARVs that cannot be used in NHP studies utilizing SIV or Env-SHIV. Indeed, this limitation motivated the development of a new class of SHIVs, termed RT-SHIVs, which were constructed by replacing the SIV *rt* gene with the NNRTI-sensitive HIV-1 *rt* gene [30]. While the relative successes of Env-SHIVs and RT-SHIVs demonstrate the utility of constructing different classes of SHIVs tailored to specific study purposes, the replicative fitness and utility of other types of SHIVs have been more disappointing [31,32], highlighting the limitations of generating SIV-based chimeric viruses for other potential viral protein targets. Limited prior experience and documentation of the natural history of infection for newly developed, study-tailored SHIVs will also significantly diminish the impact of their use. Moreover, studies intended to evaluate HIV-specific treatments, vaccinations, or immune responses targeting viral proteins other than Env or targeting more than one viral gene product cannot be pursued using existing SIV or SHIV-based NHP models. These limitations of SIV/SHIV-based models provided the impetus for the efforts to develop mmHIV-1-based NHP models of HIV-1 infection (summarized in Table 1 and Figure 1).

## 3. Obstacles to HIV-1 Infection of Macaques

The development of a modified HIV-1 capable of sustained, high-level replication in macaques first required an understanding of the obstacles for WT HIV-1 infection in these species. Unlike African NHPs, the Asian macaques (RMs, CMs, PTMs) typically used for HIV research are not naturally infected with SIV and thus when experimentally infected with the right strain of SIV they will develop pathogenic disease culminating in simian AIDS in the absence of suppressive therapy. However, HIV-1 replication is restricted in these Asian macaque species [1,33,34,35,36]. Although gibbons and chimpanzees are susceptible to HIV infection [1,2], pathogenesis is inconsistent and slow developing [37] and the protected status of these species precludes their use for HIV research.

Observed restrictions of HIV-1 replication in certain primate species led to the discovery in 2004 of species-specific antiviral restriction factors that inhibit HIV at various stages of infection. Stremlau and colleagues discovered that HIV-1 infection is potently restricted by the cytoplasmic tripartite motif-containing protein 5α (TRIM5α) restriction factor expressed in rhesus cells, while the human version of TRIM5α is far less efficient at restricting HIV-1 [38]. By contrast, SIV is far less susceptible than HIV-1 to the rhesus TRIM5α-mediated block, explaining in part why SIV can replicate to sustained high levels in rhesus macaques while HIV-1 cannot. TRIM5α binds the viral capsid (CA), blocking its function prior to viral genome integration into host-cell DNA, thereby potently inhibiting viral infection. Adaptive differences in CA sequence and structure enable SIV and HIV-1 to circumvent TRIM5α-mediated restriction in RMs and humans, respectively [38]. Similarly, the antiviral activity of another family of restriction factors, the apolipoprotein B mRNA editing enzyme catalytic subunit 3 (APOBEC3) proteins, which are packaged into virions and restrict viral replication through deamination-dependent or deamination-independent inhibition upon infection of a new target cell [39] were also found to be species specific. Although the HIV-1 Vif accessory protein significantly reduces incorporation of human APOBEC3 proteins into HIV-1 virions, thereby reducing their inhibitory effects, it fails to antagonize macaque APOBEC3 proteins [40]. Unlike HIV-1, SIV Vif proteins effectively counteract macaque APOBEC3 proteins, again indicating that the species specificity of restriction factors and the adaptive capacity of different viruses to overcome or avoid these restriction factors are key determinants of viral host range. Another restriction factor, tetherin, also known as bone marrow stromal cell antigen 2 (BST2), inhibits the release of virions produced by cells infected with a broad range of enveloped viruses by tethering budding virions to the producer cell surface, thereby limiting the ability of produced virions to find and infect new cell targets [41]. The HIV-1 Vpu protein has been shown to play a key role in counteracting human tetherin activity; however, it fails to effectively antagonize macaque tetherin [42,43,44]. Interestingly, for SIV, which lacks a *vpu* gene, tetherin can instead be antagonized by the Nef protein, highlighting the adaptive importance of restriction factor evasion for successful viral replication. The discovery and characterization of these restriction factors made it clear that productive HIV-1 infection in macaques would at a minimum require circumvention of known species-specific restriction factors.

**Table 1 viruses-16-01618-t001:** Minimally modified HIV-1 infectious molecular clones evaluated in macaques.

Virus	Backbone	Vif	CA Modifications	Env	Coreceptor Tropism	Host NHP Species	*n* *	Peak PVL (vRNA Copies/mL)	Duration of Detected PVL	Pathogenic?	Notes	Ref.
**NL-DT5R and derivatives**												
NL-DT5R	HIV-1 NL4-3	SIVmac239	SIVmac239 L4/5 swap	HIV-1 NL4-3	CXCR4	PTM	4	~10^4^	5–16 weeks	No	2 of 4 animals: anti-CD8α on days 1, 4, 7	[45]
NL-DT5R	HIV-1 NL4-3	SIVmac239	SIVmac239 L4/5 swap	HIV-1 NL4-3	CXCR4	CM	2	~10^3^	4 weeks	No		[46]
HIV-1mt ZA012	HIV-1 NL4-3	SIVmac239	NL-DT5R	HIV-1 97ZA012	CCR5	PTM	2	~10^6^	8–16 weeks	No	Serially passaged in PTM PBMC	[47]
HIV-1mt AS38	HIV-1 NL4-3	SIVmac239	NL-DT5R + SIVmac239 L6/7 swap + Q110D	SHIV MK38	CCR5	CM	2	~10^6^	10–20 weeks	No	Serially passaged in TRIMcyp homozygous CMs	[48]
HIV-1rmt MN4/ LSDQgtu	HIV-1 NL4-3	SIVmac239	HIV-1mt AS38 + M94L, R98S, G114Q	HIV-1 NL4-3	CXCR4	RM	2	~10^5^	6 weeks	No	Additional changes in pol, vpu	[49,50]
HIV-1rmt gtu+A4CI1	HIV-1 NL4-3	SIVmac239	HIV-1mtAS38 + M94L, R98S, G114Q	SHIV AD8/CI1 Recombinant	CCR5	RM	1	~10^4^	5 weeks	No	Additional changes in pol, vpu	[49,50]
**HSIV**												
HSIV-vif-NL4-3	HIV-1 NL4-3	SIVmne027	N/A	HIV-1 NL4-3	CXCR4	PTM	4	10^4^–10^5^	8–20 weeks	No	Tested in juvenile (*n* = 2) and newborn (*n* = 2) PTMs	[51]
**stHIV**												
stHIV-1	HIV-1 NL4-3	SIVmac239	N/A	SHIV KB9	CXCR4/CCR5	PTM	4	10^5^–10^6^	24 weeks	No		[52]
stHIV-A19	HIV-1 NL4-3	SIVmac239	N/A	HIV-1 AD8	CCR5	PTM	3	~10^6^	>100 weeks	No		[53,54]
stHIV-A19 (+CD8α depletion)	HIV-1 NL4-3	SIVmac239	N/A	HIV-1 AD8	CCR5	PTM	3	10^6^–10^7^	Until euthanasia	Yes		[53,54]

* Number of animals inoculated.

## 4. First-Generation Minimally Modified HIV-1s for Infection of PTMs

The growing list of host proteins expressed by macaque cells capable of restricting HIV-1 infection, and the possibility of additional such restriction factors yet to be identified, represented a daunting challenge for the development of a minimally modified version of HIV-1 that could establish high levels of viral replication and disease progression in macaques. For this reason, the first successful efforts to develop such an in vivo infection model focused first on the identification of a host primate species that would pose fewer known hurdles to HIV-1 infection.

The first clue that PTMs might represent an NHP host species with fewer limitations to HIV-1 replication came from early experiments showing that PTMs could be infected with HIV-1, albeit transiently, without sustained viral replication or evident pathogenesis or disease progression [33,55]. In an initial effort to develop a mmHIV-1 that could establish productive, chronic infection in macaques, Igarashi and coworkers evaluated a chimeric virus, designated NL-DT5R, engineered to overcome some of the known macaque restriction factors that effectively restrict HIV-1 infection. This virus was constructed by replacing the HIV-1 *vif* with the SIVmac239 *vif* to counteract macaque APOBEC3 restriction, and by replacing a 9-amino-acid stretch of the CA protein with 7 amino acids from the analogous region in the SIVmac239 CA (Figure 1), intended to counteract macaque TRIM5α [45]. Although when inoculated into PTMs this virus established a persistent population of infected cells, it measurably replicated only transiently with plasma viral load profiles comparable to WT HIV-1 inoculated into PTMs (Table 1). It was later discovered that PTMs do not express a TRIM5α protein capable of efficiently inhibiting HIV-1 infection [56], likely explaining why PTMs could support limited WT HIV-1 infection and suggesting that PTMs represented a host species with potentially at least one fewer restriction factor hurdle for mmHIV-1 to clear. This later discovery also suggested that modification of the HIV-1 CA may not have been necessary for infection in PTMs. Indeed, in initial in vitro experiments, Hatziioannou et al. found that simple substitution of the HIV-1 *vif* gene with the SIVmac239 *vif* gene in an HIV-1 NL4-3 backbone bearing a macaque adapted SHIV KB9 *env* (i.e., an HIV-1 *env* that had been adapted to support macaque infection in the context of a SHIV), but lacking any changes to CA, resulted in viruses capable of replicating in primary PTM peripheral blood mononuclear cells (PBMCs) [52]. Evidence that CA manipulations may in fact confer a fitness cost to the virus came when the authors compared this simple *vif*-only chimera with a virus containing the same *vif* swap plus replacement of the CA coding region with that of SIVmac239. When inoculated into primary PTM cells, the virus containing only *vif* replacement displayed superior replicative fitness compared with the *vif* and CA dual chimera. When intravenously inoculated into PTMs, the *vif*-only chimera, which the authors termed simian-tropic HIV-1, or stHIV-1 (Figure 1), achieved peak plasma viral loads (PVLs) that were 1–2 logs higher (10^5^–10^6^ viral RNA copies/mL) than WT HIV-1 in PTMs and, unlike WT HIV-1, maintained detectable but progressively declining PVLs through ~25 weeks of follow-up (Table 1). While these initial results were promising, there was no evident progressive CD4+ T-cell depletion or disease progression following infection with these first-generation stHIV-1 chimeras [52].

In parallel experiments, Thippeshappa and coworkers used a similar approach, distinguished by the use of a different SIV *vif*, replacing the HIV-1 NL4-3 *vif* with the *vif* coding region from SIVmne027, in an HIV-1 NL4-3 backbone that maintained the unadapted HIV-1 NL4-3 *env* [57], resulting in a virus the authors termed human-simian immunodeficiency virus (HSIV-vifNL4–3) (Figure 1) [58]. When inoculated into PTMs, HSIV-vifNL4–3 plasma viral loads reached peak levels at 2 weeks post-infection (wpi) (~10^4^–10^5^ viral RNA copies/mL), and then declined below the level of quantification within 8 to 20 wpi (Table 1) [58]. CD4+ T-cell counts in peripheral blood did not significantly decline during infection. PBMCs were PCR positive for *gag*, *vif*, *env*, and *nef* sequences through 92 wpi, suggesting that infected cells persisted in the animals for nearly 2 years [58].

The higher viral loads observed in PTMs infected with stHIV-1 compared with PTMs infected with HSIV-vifNL4–3 (Table 1) suggested that while Vif-mediated APOBEC3 evasion was critical for HIV-1 replication in PTMs, choice of *env* gene may also be important. Although the initial stHIV-1 construct and HSIV-vifNL4–3 were similarly derived by replacing the *vif* gene region in the HIV-1 NL4-3 backbone with an SIV *vif*, stHIV-1 also incorporated a macaque-adapted HIV-1 *env* cloned from SHIV-KB9 [59] whereas HSIV-vifNL4–3 maintained the unmodified HIV-1 NL4-3 *env* [51] (Figure 1). Efforts to develop SHIVs bearing a variety of Envs have shown that most WT HIV-1 Envs cannot efficiently utilize macaque CD4 for viral entry, though rare HIV-1 Envs capable of engaging macaque CD4 without adaptation and alteration have been identified [28,60,61,62,63]. Historically, within the context of SHIV development, this limitation has been overcome through animal-to-animal serial virus passage to drive the acquisition of adaptive mutations that improve macaque CD4 binding, as was the case for the SHIV-KB9 *env*. More recently, specific point mutations that enhance macaque CD4 engagement for most HIV-1 Envs, and which may be of use for next-generation mmHIV-1 constructs, have been identified [28,29]. The inability of most unadapted HIV-1 Envs to effectively mediate infection of cells expressing macaque CD4 poses a potential limitation for future applications of mmHIV-1 models, where the incorporation a variety of HIV-1 Envs with specific features of interest would be useful. At a minimum most HIV-1 Envs will likely require the incorporation of point mutations that enhance macaque CD4 engagement, though the potential impact of these point mutations on the specific Env phenotypes of interest will have to be evaluated on a case-by-case basis.

Irrespective of their capacity to bind macaque CD4, the Envs incorporated into each of the first-generation mmHIV-1 constructs utilized C-X-C motif chemokine receptor 4 (CXCR4) as an entry coreceptor, either exclusively or in addition to C chemokine receptor type 5 (CCR5). Subsequent mmHIV-1 iterations shifted to the use of Envs with strict CCR5-tropism to more authentically recapitulate the phenotype of transmitted HIV-1 (Figure 1). While the original NL-DT5R clone encoded the CXCR4-tropic HIV-1 NL4-3 Env, a second-generation construct, designated HIV-1mtZA012, was developed by swapping the *env* gene from the CCR5-tropic HIV-1 97ZA012 into the NL-DT5R construct. Following serial passage of HIV-1mtZA012 in PTM PBMCs, the resultant virus replicated to peak plasma viral loads that were more than 10-fold greater in PTMs than had been reported for NL-DT5R (Table 1), suggesting the potential importance of CCR5-tropism on acute viral replicative fitness. However, like the first-generation mmHIV-1s, HIV-1mtZA012 viral loads declined below quantification limits by 8–16 wpi [47]. As discussed below, further efforts to develop mmHIV-1-based NHP models of HIV-1 infection have often similarly incorporated HIV-1 Envs with CCR5 tropism.

While these initial efforts to develop mmHIV-1 for use in NHPs models focused on the use of southern PTMs due to their capacity to support limited WT HIV-1 replication, similar susceptibility to WT HIV-1 infection was also recently shown for northern pigtail macaques (NPMs; *Macaca leonina*), a separate species of macaque that until recently were considered a subspecies of southern PTMs (*Macaca nemestrina*). Like southern PTMs, NPMs also do not express a TRIM5α protein that restricts HIV-1. In a recent report, four NPMs were successfully infected with WT HIV-1 NL4-3, with peak plasma viral loads reaching 10^4^–10^5^ viral RNA copies/mL by 1 wpi, but then decreasing rapidly to levels below assay quantification limits over the following weeks, akin to what had been previously observed for WT HIV-1 infection in southern PTMs [64]. Despite the rapid decline in PVLs, peripheral blood and lymphoid organs showed cell-associated viral DNA and RNA that persisted for about 3 years and could be reactivated with latency reversing agents suggesting the establishment of a persistent viral reservoir in HIV-1-infected NPMs [64]. While the establishment of long-lived sources of replication competent virus may also occur following WT HIV-1 infection of southern PTMs, it has not yet been evaluated or reported, and the utility of a model of persistent reservoir establishment in the setting of robust virologic control and/or limited replication and pathogenesis remains unclear. These findings suggested that NPMs may also be attractive host species to consider for further development of mmHIV-1-based NHP models of HIV-1 infection.

## 5. Further Development of PTM-Based Models

Although molecularly engineered modifications to enable HIV-1 to circumvent macaque APOBEC3 restriction and perhaps engage the macaque CD4 receptor more effectively led to both increased acute viral replication levels and longer periods of detectable plasma viremia in PTMs, it was clear that additional adaptative changes would be required for mmHIV-1 to achieve sustained, elevated plasma viral loads and AIDS-like viral pathogenesis. In an effort to derive a pathogenic mmHIV-1 with improved replication capacity, we and others utilized animal-to-animal serial passages, an approach previously used to drive the acquisition of adaptive mutations that improved the replicative fitness of chimeric SHIVs. In our work, we started with the original stHIV-1 construct, composed of the HIV-1 NL4-3 viral backbone encoding the SIVmac239 Vif, and generated four variants by replacing the dual-tropic (i.e., capable of using either CCR5 or CXCR4) KB9 *env* with four unique CCR5-tropic HIV-1 *envs*: YU2, BaL, AD8, and a modified version of the KB9 Env that was made CCR5 tropic through the replacement of the Env V3 loop with that of AD8 (KB9(AD8)). All four of these CCR5-tropic stHIV-1s were pooled and intravenously inoculated into two animals, which served as the donors for the first of several animal-to-animal serial passages. Intriguingly, in these first infected animals, virus bearing the AD8 *env* predominated the replicating virus population early after infection and thereafter, including in subsequent passage recipients. Separate, later work identified AD8 as a rare HIV-1 Env capable of effectively utilizing macaque CD4 without the need for adaptation or modification, again underscoring a role for Env in contributing to macaque tropism. Acute plasma viremia reached ~10^5^ vRNA copies/mL in these initial stHIV-1-infected animals and declined thereafter [53]. Virus was subsequently serially passaged three times in PTMs with experimental antibody-mediated depletion of CD8α+ cells, intended to boost viral replication levels and associated diversity, in all passage recipients at the time of inoculation. One of three recipients of the third serial passage displayed sustained plasma viral loads >10^6^ viral RNA copies/mL for 28 weeks, with progressive loss of CD4+ T cells in blood and the development of an AIDS-defining extranodal B-cell lymphoma. Virus from this animal was passaged into four additional recipients, two of which received cell-depleting CD8α+ antibody at the time of inoculation, while two received virus alone without antibody administration. Peak viremia reached ~10^7^ viral RNA copies/mL, with CD4+ T-cell depletion in blood and gut of CD8α-depleted PTMs and the development of AIDS-defining disease, whereas in the non-depleted animals the level of acute viremia was ~10^6^ viral RNA copies/mL with progressive control of viral replication thereafter. This pattern was maintained following one further passage from a CD8α-depleted animal into two more animals, one of which was CD8α depleted while the other was not, with CD8α depletion again resulting in pathogenic stHIV-1 infection characterized by high chronic viral loads and progressive CD4 loss, while without CD8α depletion stHIV-1 replication gradually declined to low levels.

When we evaluated the adaptive mutations acquired by stHIV-1 over the course of serial passage in PTMs, viruses associated with elevated viral loads and pathogenesis (in CD8α depleted animals) contained mutations that conferred the ability to circumvent additional restriction factors. Mutations were acquired and became fixed in the Vpu of stHIV-1 population, conferring an improved ability to antagonize PTM tetherin [53]. Additionally, when PTM-adapted stHIV-1 infectious molecular viral clones were isolated from animals with progressive disease, several different mutations in the CA coding region were identified that rendered the virus less susceptible to the more recently discovered MX dynamin-like GTPase 2 (Mx2) restriction factor, which inhibits HIV replication by preventing nuclear import of the viral pre-integration complex [54]. Several different mutant CA genotypes were identified, displaying a range in Mx2 sensitivity. Intriguingly, CA mutations that conferred the highest degree of Mx2 resistance also conferred a pronounced replication defect, but CA mutations that conferred only partial Mx2 resistance did not impact viral replicative fitness, again highlighting the vulnerability of the virus to CA alterations. After screening a number of infectious molecular clones isolated from passage recipients that were CD8α-depleted and presented a pathogenic infection course, a molecular clone designated stHIV-A19 (Figure 1), derived from the first animal that progressed to clinical AIDS, was selected for future work. This infectious molecular clone contained the HIV-1 AD8 *env*, mutations in Vpu that conferred improved activity against PTM tetherin, and CA mutations associated with intermediate PTM Mx2 resistance, and like the passaged viral swarm from which it was derived, stHIV-A19 consistently caused AIDS (marked by CD4+ T-cell depletion and the development of AIDS-defining clinical diseases) in CD8α+ cell-depleted PTMs. However, also like the passaged virus swarm from which it was derived, in immunocompetent PTMs (i.e., those not CD8α-depleted at the time of infection), stHIV-A19 viral loads progressively declined during the chronic phase of infection with no progressive CD4+ T-cell loss or disease progression (Table 1), indicating a requirement for CD8α+ cell-depletion for disease development in this model [54].

Animal-to-animal serial passaging of HSIV-vifNL4-3 was also pursued by the Kimata group. As noted above, SIV genomes encode the Vpx accessory protein, which counteracts the SAM and HD domain containing deoxynucleoside triphosphate triphosphohydrolase 1 (SAMHD1) restriction factor and is required for SIV pathogenesis in macaques [65,66,67]. By contrast, HIV-1 genomes do not include a *vpx* gene and because the *vpx* open reading frame (ORF) overlaps the *vif* ORF in the SIV genome, a partial *vpx* gene fragment is typically introduced along with the SIV *vif* upstream of *vpr* during mmHIV-1 construction. Studies by the Kimata lab showed that the presence of this *vpx* gene fragment resulted in a Vpr expression defect in HSIV-vifNL4-3 [68], suggesting that defective Vpr expression or the lack of a functional *vpx* gene may contribute to the lack of viral pathogenesis for HSIV and other mmHIV-1s. To address these possibilities, Thippeshappa and colleagues generated infectious HSIV-vif clones in which either Vpr expression had been restored or the full SIV *vpx* gene had been incorporated into HSIV-vif backbones that also incorporated CCR5-tropic HIV-1 *envs* (YU2 or AD8) [68]. Using a pooled inoculum strategy, these new CCR5-tropic Vpr- or Vpx- expressing clones were intravenously co-inoculated into two naïve PTMs along with two CXCR4-tropic HSIV-vifNL4-3 clones derived from prior animals that had been infected for ~4 years. Virus was subsequently serially passaged from these two animals into a third animal, and then from that third animal into a fourth animal, without the use of CD8α depletion in either the passage virus donors or recipients, with no evident improvements in viral replication levels or observed pathogenesis. Following two rounds of virus serial passage, 54 full-length viral genomes were cloned from the last animal in the passage series and evaluated for their capacity to produce infectious virus. Three of the PTM-adapted viral genome clones produced infectious virus of adequate titer for subsequent evaluation. Intriguingly, all three clones were genetically similar to the two CXCR4-tropic HSIV-vifNL4-3 clones in the initial viral inoculum that had been derived from a long-term infected animal. However, through viral recombination in vivo, these clones acquired mutations present in the virus inoculum pool that restored HIV-1 Vpr expression, suggesting the importance of Vpr function for in vivo pathogenesis and underscoring the challenges of constructing chimeric viral genomes from viruses with disparate overlapping ORF organization [68]. It is unclear if a similar defect in Vpr expression occurs in the stHIV-1 viruses, which were constructed using a different SIV *vif* (from SIVmac239 rather than SIVmne027), though of note there were no mutations selected for in the stHIV-1 *vpr* gene following multiple animal-to-animal serial passages.

## 6. Strengths and Utility of PTM-Based Minimally Modified HIV-1 Models

### 6.1. Genetic Similarities between HIV-1 and mmHIV-1

Genetic differences between HIV-1 and SIV can have both predictable and unpredictable impacts on viral biology. Differences in immunodominant cytotoxic T-lymphocytes (CTL) epitopes, and differences in Env and other viral proteins may lead to qualitative differences in host immune responses to infection and/or immunization. One of the strengths of mmHIV-1 models is that the viral genomes described thus far comprise 88–94% HIV-1 genome sequence. HIV vaccine approaches focused on prevention of virus acquisition (i.e., “sterilizing immunity”) targeting one or multiple viral gene products can theoretically be directly tested using these models, which readily establish infection with acute plasma viremia following inoculation. Because each of the macaque tropic HIV-1 strains we have described here replicate relatively well during the acute phase of infection and share extensive genetic identity with HIV-1 suggests that understanding the mechanisms underlying progressive viral control displayed in PTMs could be highly informative and relevant for efforts to develop HIV-1 vaccination and treatment approaches. In addition, though mmHIV-1 replication is spontaneously controlled, persistent sources of replication competent virus are established in these models, suggesting the establishment of populations of long-lived latently infected cells. While additional characterization of viral persistence in animals infected with macaque-tropic HIV is needed, these models may also prove useful for the evaluation of HIV-specific interventions intended to target latently HIV-1-infected cells in vivo.

A nonhuman primate model of HIV-1 infection based on the use of a mmHIV-1 also enables in vivo studies focused on aspects of HIV-1 biology and pathogenesis that are not faithfully recapitulated by SIV-based models. For example, during the in vivo serial passaging of stHIV-1 bearing an initially CCR5-tropic AD8 Env, a CXCR4-tropic variant, bearing a four amino-acid deletion in the Env V3 loop, emerged in one passage recipient in association with a precipitous decline in CD4+ T cells in blood and end stage disease progression [53]. Like stHIV-1 AD8 and virtually all transmitted HIV-1 variants [69,70,71], the pathogenic SIV strains used in NHP models of HIV-1 infection utilize CCR5 as their primary entry coreceptor. However, while the emergence of CXCR4-tropic viral variants (i.e., “coreceptor switching”) is relatively commonplace during untreated HIV-1 infection in humans, typically in association with dramatic CD4+ T-cell loss and the transition to end stage disease [69,70,71,72,73], the emergence of SIVs capable of using CXCR4 during late-stage disease has not been reported.

### 6.2. Evaluations of Pre-Exposure Prophylaxis (PrEP) Agents

Although in the absence of experimental immune modulation existing mmHIV-1 strains do not maintain elevated plasma viral loads during the chronic phase of infection, their ability to establish relatively high levels of acute viral replication has indicated their potential utility to evaluate PrEP with ARVs, particularly those with reduced potency against SIV. In a recent proof-of-concept experiment, we showed that the long-acting CA-targeted ARV lenacapavir (LEN) protected PTMs from infection with a stringent, high-dose intravenous challenge of stHIV-A19 [74]. Importantly, when we compared LEN potency against HIV-1 NL4-3, stHIV-A19, and SIVmac239, we found that stHIV-A19 and HIV-1 NL4-3 were comparably sensitive to LEN, consistent with our expectations given that stHIV-A19 encodes an HIV-1 CA; however, SIVmac239 was approximately 6-fold less sensitive to LEN [74], falling well outside the range in sensitivity to LEN reported for circulating HIV-1 variants [75]. This difference between SIV and HIV-1 in sensitivity to an ARV is not unique to LEN, as noted above for NNRTIs [30,76]. Presumably, because each of the mmHIV-1 constructs described to date possesses an unaltered HIV-1 reverse transcriptase coding region as well as HIV-1 sequence throughout most or all of the rest of the viral genome other than *vif*, sensitivity against NNRTIs and other drug classes will likely be very similar to circulating HIV-1 strains, though such comparable sensitivity should be demonstrated for each agent proposed for in vivo testing in mmHIV-1 models.

### 6.3. Utility of PTMs as a Host Species for Modeling for HIV Infection

Thus far, efforts to establish a mmHIV-1 model in RMs have been largely disappointing (discussed below) [49,50,77] so the application of mmHIV-1 in NHP models is effectively limited to studies that utilize southern or northern PTMs. While restriction to PTM species can pose some limitations, there are some advantages to the use of PTMs. First, apart from likely expressing fewer HIV inhibitory restriction factors than their rhesus counterparts, PTMs may also be more likely than RMs to develop a pathogenic disease course following mmHIV-1 infection due to a general proclivity toward elevated lentiviral replication levels and pathogenesis. Prior studies comparing SIV infection in PTMs and RMs have shown that PTMs often display elevated levels of immune activation prior to viral infection, with associated higher levels of SIV viral replication and more rapid disease development following infection, likely due at least in part to poorer gut barrier integrity and higher levels of microbial translocation in PTMs [78]. In addition, because PTMs are larger on average than RMs, often larger specimen volumes may be obtained. Finally, germane to the potential use of current mmHIV-1 models for evaluation of infection prevention modalities, female PTMs are of particular use for intravaginal challenge experiments because they exhibit a 32-day lunar menstrual cycle similar to humans [79]. The relevance of PTM vaginal physiology has allowed researchers to assess menstrual phase-dependent susceptibility to HIV with a level of experimental control not achievable in humans [80,81].

## 7. Limitations of PTM-Based Models

While the use of PTMs for HIV research may have some advantages, there are also several limitations associated with the use of these animals. First, PTMs are less readily available for research use compared with RMs and may be more difficult and/or costly to obtain. Additionally, the availability of PTM-specific reagents, such as antibodies and analytical kits is more limited than for RMs or CMs, and the extent of genetic and immunological characterization, including the markers and functions of various immune cell subsets and major histocompatibility complex (MHC) genotyping information, is far less extensive for PTMs compared with RMs. Finally, the larger size of PTMs compared to RMs can pose practical/logistical challenges for housing and maintenance costs.

## 8. Efforts to Develop Minimally Modified HIV-1 for Infection of CM and RM

Much of the work to develop mmHIV-1 models of HIV infection in NHPs has focused on the use of PTMs (northern or southern) because they pose a “path of least resistance” for the expansion of HIV-1 tropism into macaques. But given some of the challenges and limitations of working with PTMs, there have been some reported attempts to develop mmHIV-1 capable of infecting CMs and RMS. Unlike PTMs, both CMs and RMs express HIV-inhibitory TRIM proteins in addition to HIV-inhibitory APOBEC3G and other restriction factors, and thus have an additional restriction factor barrier for mmHIV-1 to overcome. In ongoing efforts to develop viruses better able to circumvent CM TRIM restriction, a number of studies have iteratively modified NL-DT5R (Figure 1), which contains both an SIV *vif* and the incorporation of a small region of the SIV CA coding region, through the addition of further CA modifications, the incorporation of CCR5-tropic Envs, and serial passage in simian cell lines [47,82]. Notably, these studies revealed that the potency of HIV-1 inhibition by CM TRIM is dependent on the TRIM genotype of the CM target cells. Although CM TRIM5α potently restricts mmHIV-1, some CMs possess one or two TRIM alleles that express a TRIM-cyclophilin A fusion protein (TRIMcyp) rather than TRIM5α. Cells from rare animals homozygous for the TRIMcyp genotype displayed greater permissivity for in vitro mmHIV-1 infection [83]. Through sequential modification and adaptation of NL-DT5R and exclusive use of CM animals homozygous for TRIMcyp, modest improvements to mmHIV-1 infection of CM animals have been made. When inoculated into CMs, the parental NL-DT5R virus established transient, low-level infection with viral loads that peaked at ~10^3^ viral RNA copies/mL plasma but rapidly declined below assay detection limits by week 4 post-infection (Table 1) [46]. More recently, Ode and colleagues serially passaged an extensively modified derivative of NL-DT5R, designated HIV-1mtAS38, in CM animals homozygous for TRIMcyp (Figure 1). Though peak plasma viral loads in the final passage recipients reached ~10^6^ vRNA copies/mL, viral loads declined below assay detection limits within 10–20 weeks of infection (Table 1) [48]. These extensive efforts highlight the nuanced challenges of mmHIV-1 development in macaque species that express TRIM proteins capable of potent HIV-1 inhibition.

Rhesus macaques have proven to be the most difficult macaque species in which to derive a mmHIV-1 capable of productive infection. Though NL-DT5R could replicate transiently in PTM and CM animals, and in primary CM cells without any further adaptation, it was unable to replicate measurably in primary RM cells [77]. To derive a modified HIV-1 capable of replication in RMs, Nomaguchi and coworkers iteratively incorporated several additional changes into the NL-DT5R genome (Figure 1), including multiple modifications to the CA coding region, two changes in the integrase coding region of *pol*, and replacement of the transmembrane domain of *vpu* with the corresponding region from SIVgsn166 (SIV isolated from the greater spot-nosed monkey) to counteract RM tetherin [49,50]. Viruses constructed in this fashion and including either a CXCR4-tropic or CCR5-tropic Env, collectively termed rhesus macaque tropic HIV-1 (HIV-1rmt), were able to replicate in RM PBMC in vitro, suggesting the critical importance of overcoming TRIM5α and tetherin restriction to replicate in RM cells. When these clones were inoculated into RMs, they were able to establish acute plasma viremia, peaking between 10^4^–10^5^ viral RNA copies/mL, but by 6 wpi plasma viral loads in all animals declined below assay detection limits (Table 1) [49]. These efforts illustrate that expansion of HIV-1 species tropism to RMs poses a much more substantial barrier than PTMs or even CMs and that further work to understand the full extent of these barriers will be required to derive a mmHIV-1 capable of sustained viral replication with disease progression in RMs, and that such a virus may necessarily be more significantly altered (i.e., less HIV-like) than would be necessary for PTM hosts.

## 9. Minimally Modified HIV-1 Models: Future Directions for the Field

### 9.1. Development of a Pathogenic mmHIV-1

It remains unclear why plasma viral loads invariably progressively decline to levels near or below assay quantification limits following inoculation of any of the available mmHIV-1 strains into immunocompetent PTMs, but the impact of CD8α antibody administration may provide some clues. While CD8α antibody administration during chronic infection has been shown in several mmHIV-1 models to result in transient increases in plasma viremia [45,46,53], we have shown that when implemented at the time of infection with the PTM-adapted stHIV-A19 CD8α antibody administration results in an infection course characterized by sustained elevated plasma viral loads, CD4+ T-cell loss, and progression to AIDS-defining clinical diseases [54]. Several different possible mechanisms could underlie this effect. The most obvious possibility is that because CD8α antibody administration results in the profound depletion of CD8+ T cells, including CD8+ CTLs, adaptive CTL-mediated immune responses that are typically able to progressively control mmHIV-1 in PTMs are disrupted. However, if CTL responses are indeed the primary mediator of mmHIV-1 control, it would suggest a surprising capacity for outbred PTMs to uniformly mount similarly effective CTL responses against HIV-1, regardless of their MHC genotypes, and that mmHIV-1 is for some reason unable to develop escape mutations to these CTL responses. An alternative or additional possibility is that NK cells, which are also profoundly depleted by CD8α antibody administration due their high surface expression of CD8α in macaques, play a major role in controlling HIV-1 infection in PTMs.

CD8α is also expressed on the surface of some subsets of plasmacytoid dendritic cells (pDCs), the major in vivo producers of type I interferon (IFN). Type I IFNs are upregulated during HIV-1 and SIV infections and are thought to impart some degree of control on HIV-1 replication in humans [84,85,86,87]. It is possible that in macaques mmHIV-1 strains are exquisitely sensitive to type I IFN-mediated inhibition [88,89], including the upregulation of IFN-stimulated restriction factors. Though actual depletion of pDCs following CD8α administration has not been shown, CD8α antibody administration may lead to diminished Type I IFN production, through either the depletion or modulation of pDCs.

Recent studies conducted in NPMs have strengthened the notion that virus-specific regulation of innate inflammatory responses in macaques compared with humans may partially explain why mmHIV-1s fail to establish pathogenic infection in immunologically intact animals, though the exact mechanisms remain poorly defined. In human cells, increased expression of proinflammatory cytokines such as tumor necrosis factor alpha (TNF-α), interleukin 6 (IL-6), and interleukin-12, which contribute to HIV-induced immune activation, can be triggered by Toll-like receptor (TLR) 8 sensing of HIV-1 infection [90]. In a series of in vitro and ex vivo experiments, Pang and colleagues found that NPM TLR8 was less robustly responsive to stimulation by TLR8 agonist drugs or inactivated stHIV-1 or SIV particles than human or rhesus TLR8 proteins [91], suggesting a potential mechanism for limited immune activation and inflammation in mmHIV-1-infected NPMs. This phenotype is conceptually similar to that of the sooty mangabey, a natural host for SIV that notably does not develop AIDS following SIV infection, with associated lower levels of immune activation than observed in SIV-infected RMs or people with HIV-1 [92]. However, in a separate study, the Pang group showed that despite substantially lower viral burden, HIV-1 infection induced more robust increases in TNF-α and IL-6 mRNA expression than did SIVmac239 infection in NPMs, highlighting the challenges of analyzing the complex and often redundant regulation of innate immune activation. In addition to suggesting a potential role for differential inflammatory responses in mmHIV-1-infected NPMs, He et al. also found that IFNα and a number of downstream type I interferon-stimulated genes (ISGs) [93] were more dramatically upregulated in NPMs infected with WT HIV-1 or stHIV-1 than with SIVmac239 [93]. Intriguingly, among the ISGs found to be more robustly induced by HIV-1 infection than SIVmac239 infection in NPMs was interferon alpha inducible protein 27 (IFI27), which was shown to more potently inhibit HIV-1 infection than human IFI27 [91]. Whether these observations extend to southern PTMs or are specific to NPMs, and whether they can be overcome through targeted manipulation of the viral genome or other approaches remains to be determined.

### 9.2. Demonstration of mmHIV-1 Mucosal Transmissibility

The overwhelming majority of HIV-1 transmissions in humans occur via sexual contact, with viral transmission occurring across a mucosal barrier [94,95,96,97,98,99,100]. However, for each of the mmHIV-1 models, all of the macaque infections reported thus far have occurred via the intravenous route of exposure. While this route maximizes the efficiency of virus exposure and minimizes the variability between inoculations, demonstration of mucosal virus transmissibility will be critical to extend the utility of mmHIV-1 models to pre-clinical evaluations of prevention strategies that employ clinically relevant routes of virus exposure, including preventive vaccine evaluations, ARV and antibody-based PrEP evaluations, and microbicide assessments. Given the relevance of PTM menstrual cycles and vaginal biology relative to humans, demonstration of intravaginal transmission will be particularly useful for further development of these models.

### 9.3. Demonstration of the Establishment of a Rebound-Competent Viral Reservoir in mmHIV-1-Infected Macaques

While antiretroviral therapy (ART)-mediated suppression of viral replication to clinically relevant levels and the establishment of persistent viral reservoirs have been demonstrated for a number of SIV- and SHIV-based NHP models, these models are limited in their capacity to facilitate the evaluation of proposed HIV functional cure or eradication strategies with specificity for HIV or markedly different potency against SIV/SHIVs, such as therapeutic vaccinations targeting viral targets other than Env. Though each of the reported mmHIV-1 strains described thus far are spontaneously controlled in immunocompetent macaques, perhaps precluding evaluation of ART mediated suppression or the demonstration of off-ART rebound definitively indicative of viral reservoir establishment, an assessment of long-term ART and subsequent cessation in stHIV-A19 (or other mmHIV-1)-infected animals CD8α depleted at the time of virus inoculation may be useful. Because CD8α depleted PTMs infected with stHIV-A19 maintain elevated plasma viral loads in the absence of ART, such animals would enable evaluation of ART regimens to achieve virologic suppression of mmHIV-1 in PTMs and assessments of off-ART viral rebound. Such assessments should include monitoring for CD8+ T-cell and NK-cell recovery during ART, as such recovery would be critical if current models are to be used to evaluate immunization or immune modulation approaches to enhance viral reservoir elimination or control.

### 9.4. Characterization of Residual Infected Cells That Persist in mmHIV-1 Models

A substantial portion of the overall effort to identify an HIV-1 eradication strategy is focused on identifying latency reversing agents (LRAs) capable of safely and robustly reactivating residual latent viral genomes in vivo so that persistent infected cells can be targeted for elimination by immunologic or other means. Despite spontaneous control of plasma viral loads to levels that are typically near or below viral load assay quantification limits, several studies have reported the establishment of a relatively long-lived population of infected cells, evidenced by the persistence of cells harboring either viral DNA or replication competent virus, in mmHIV-1-infected macaques not receiving ART [58,64,68]. The presence of these persistent infected cells in the absence of detectable plasma viremia is suggestive of a population of latently infected cells, perhaps akin to that which persists in people with HIV on suppressive ART. Further characterization of these persistent infected cells, including their phenotypes, memory status, and activation state, and the viral genomes they harbor, including intactness, clonality, and magnitude and extent of spontaneous gene expression, in mmHIV-1-infected macaques, could provide important information regarding the potential utility of mmHIV-1 models to evaluate the activity and safety of latency reversing agents or other approaches for targeting the rebound competent viral reservoir in vivo. Should these residual infected cells reflect key aspects of latently infected cells that persist in humans with HIV-1 on ART, mmHIV-1 models may provide a simple, ART-free in vivo experimental system in which proposed LRAs may be assessed.

## 10. Concluding Remarks

Through the parallel efforts of a small number of laboratories, several similar but distinct mmHIV-1-based NHP models of HIV-1 infection have been developed. While each of these models are characterized by variable but reasonably high levels of acute viral replication following intravenous inoculation into macaques, plasma viral loads spontaneously decline in all cases to levels that are low or below the level of detection during the chronic phase of infection. As a result, the current models are useful for some applications, in particular the evaluation of infection prevention approaches or studies focused on illuminating the factors that define host-species tropism for primate lentiviruses, but not for many others. Unlocking the potential of these models, which could revolutionize the utility and power of NHP models of HIV-1 infection, will require the development of a mmHIV-1 capable of establishing a pathogenic infection course characterized by sustained, elevated plasma viral loads, progressive CD4+ T-cell loss, and the development of AIDS-defining clinical endpoints. In our own work, we have derived an adapted mmHIV-1 infectious molecular clone, stHIV-A19, capable of recapitulating each these key features of HIV-1 infection, but only in animals that receive cell-depleting CD8α antibody administrations at the time of virus inoculation. Understanding the mechanisms that underlie this requirement for CD8α+ cell depletion at the time of infection will be a critical step toward the development of a pathogenic mmHIV-1 model of HIV-1 infection.

## Figures and Tables

**Figure 1 viruses-16-01618-f001:**
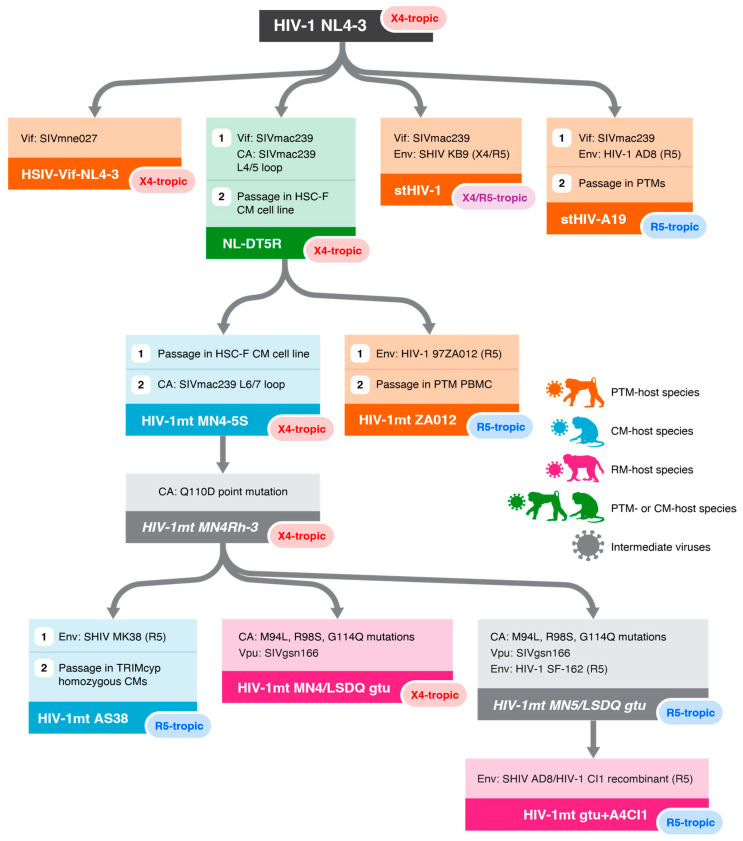
Derivation pathway of minimally modified HIV-1 infectious molecular clones that have been evaluated in nonhuman primates (NHPs). Color coding corresponds to the NHP species in which each virus has been inoculated. X4-tropic, CXCR4-tropic; R5-tropic, CCR5 tropic; X4/R5-tropic, CXCR4/CCR5 dual tropic; Env, envelope; CA, capsid; Vif, viral infectivity factor; Vpu, virus protein u; PTM, pigtail macaque; CM, cynomolgus macaque; RM, rhesus macaque.

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
