# Peer review of "Minimally Modified HIV-1 Infection of Macaques: Development, Utility, and Limitations of Current Models"

_viruses, 2024, doi:10.3390/v16101618_

Round 1
Reviewer 1 Report
Comments and Suggestions for Authors
This review of the difficulties and complexities of optimizing an animal model for HIV/AIDS is a pleasure to read: clear, informative, and well-organized.
There was only one point at which I was uncertain of the intended meaning: I don’t understand the sentence on lines 336-339. Part of the difficulty is I don’t know what they are referring to as “the rare TRIMcyp genotype”, but I think that is not the entire problem and I think the whole sentence should probably be rewritten, perhaps expanded if necessary.
Author Response
Reviewer 1 found that our manuscript was a “pleasure to read: clear, informative, and well-organized.” We thank the reviewer for these positive comments. Reviewer 1 had one minor comments.
Comment 1: There was only one point at which I was uncertain of the intended meaning: I don’t understand the sentence on lines 336-339. Part of the difficulty is I don’t know what they are referring to as “the rare TRIMcyp genotype”, but I think that is not the entire problem and I think the whole sentence should probably be rewritten, perhaps expanded if necessary.
- Response 1: We thank the reviewer for this helpful comment. We have rewritten this section of the manuscript (lines 344-349) to more clearly explain this point about TRIM genotypes in cynomolgus macaques.
Reviewer 2 Report
Comments and Suggestions for Authors
The manuscript by Sharma et al provides an excellent review of the development and possible applications of mmHIV-1.
Comments
1. In the review it is discussed that not all HIV-1 env sequences may be equally replication competent when incorporated in a mmHIV-1. This might put some limitations on the model with regard to HIV vaccine evaluation. This item could be discussed as one of the possible limitations.
2. In line 393 the word “role” is lacking.
Author Response
Reviewer 2 noted that the manuscript “provides and excellent review of the development and possible applications of mmHIV-1”. We are gratified that the reviewer had a positive opinion of the manuscript. Reviewer 2 had two minor comments.
Comment 1: In the review it is discussed that not all HIV-1 env sequences may be equally replication competent when incorporated in a mmHIV-1. This might put some limitations on the model with regard to HIV vaccine evaluation. This item could be discussed as one of the possible limitations.
- Response 1: We thank the reviewer for this suggested addition to the text. We have added lines 107-114 on pages 7 and 8 of the current draft of the manuscript to discuss this point.
Comment 2: In line 393 the word “role” is lacking.
- Response 2: We have added the missing word “role” to the sentence (now line 411 in the current draft)